# The Struggle between Cash and Electronic Payments

**Zsófia Pintér [1], Mónika Zita Nagy [1,*], Katalin Tóth [2] and József Varga [3,4,5]**

1   Insstitute of Agriculture and Food Economics, Hungarian University of Agriculture and Life Sciences, H-7400 Kaposvar, Hungary
2   "Böhönyei Szabadság" Agricultural Co., Böhönye, H-8719 Marcali, Hungary
3   Institute for Rural Development and Sustainable Economy, Hungarian University of Agriculture and Life Sciences, H-7400 Kaposvar, Hungary
4   Department of Finance, Corvinus University of Budapest, H-1093 Budapest, Hungary
5   Faculty of Economics, Socio-Human Sciences and Engineering, Sapientia Hungarian University of Transylvania, RO530104 Miercurea Ciuc, Romania
*   Correspondence: nagy.monika.zita@uni-mate.hu

**Abstract:** The assessment of consumer behavior regarding the choice of financial instruments may be extremely important in the near future, since the fight between cash and electronic money has reached a turning point, and electronic payments are slowly defeating cash. On one hand, in the long term, this possible separation threatens sustainable development goals, and on the other hand, financial awareness can affect the number of purchases and savings. In a survey of 499 people, we examined the reasons behind their decisions, with a particular focus on financial awareness. The result shows that the vast majority of Hungarian consumers are not yet ready to fully accept electronic payments. It can be stated that financial awareness is not present at all in one-fifth of respondents, and one-third are influenced by habituation in everyday shopping situations, which indicates a lack of financial awareness. Based on our results, we have concluded that our consumers still find it difficult to abandon cash payments. Financial awareness needs to be improved in parallel with the Hungarian government's strategy to reduce the use of cash.

**Keywords:** cashless society; waste; sustainability; means of payment; consumer survey; digitalization; financial awareness

## 1. Introduction

In our publication, we analyze the topic of electronic and cash payments. As technological innovations and the development of financial instruments come to the forefront, consumers who do not take advantage of the opportunities offered by digitalization may break away from the group. In the long term, this endangers the sustainability goals, on the other hand, financial awareness can affect the number of purchases and savings.

The relevance of our study is given by the fact that the fight between cash and electronic money has reached a turning point in Hungary, as electronic payments have defeated cash by the end of 2020 (Hergár 2021), and by the end of 2021, the total value of card purchases (HUF 10.7 thousand billion) is approximated to exceeded card cash withdrawals by 25% (HUF 8.6 thousand billion) (Hergár 2022). This process is supported by the Hungarian Banking Association with active partnership and formulation of proposals. The Government of Hungary has also prepared a strategy, one of the goals of which is to improve financial awareness and reduce the use of cash (Bálint et al. 2020). In addition, the authors consider it important to assess consumers' readiness for full acceptance of electronic payments. We surveyed 499 people, based on the answers we received in our previous article, the effect of the pain of payment (Pintér et al. 2021). In this article, we analyze the full acceptance of electronic payment through consumer habits.

In this review of the literature, digitization and the financial habits of consumers are reviewed from the point of view of sustainability, and the results of similar surveys are also

summarized. After stating the hypotheses established from the literature, the results of our own questionnaire are detailed. Based on our results, we came to the conclusion that our consumers still find it difficult to abandon cash payments.

## 2. Literature Review

### 2.1. Is Adaptation to Digitalization Key to Sustainability?

Consumers in Western societies (in the United States of America and the countries of the European Union) can observe a change in their consumption values because it is where environmental awareness comes to the forefront (Brávácz 2015). Awareness has two effects: on one hand, it directly influences sustainable consumption behavior, and on the other hand, it has an indirect positive effect on environmental concerns (Helm and Subramaniam 2019).

At the same time, the effects of digitalization are now indisputably visible to everyone, as technologies based on it are appearing everywhere and are slowly replacing our analog lives. These changes result in new consumer habits and allow access to new, large amounts of information about consumers. This contributes to the understanding of consumers and their motivation, e.g., understanding user experiences in e-commerce can be a key e-business aspect (Fedushko and Ustyianovych 2022). However, scientific research related to digitalization and its assessment is still in its initial phase (Brenner and Hartl 2021).

Changes can also be observed in the field of financial instruments used for payment, but not all consumers want or are able to use them. In many cases, Hungary is still unable to benefit from the advantages of digitalization, because the development of the necessary competencies does not appear in education (Éltető 2021), and digital competence would also require a technical and practice-oriented approach (Godhe 2019).

These are also important issues in social terms since the lack of application of digitalization can mean a disconnection. In 2015, the UN set the Sustainable Development Goals (SDGs) (Mondejar et al. 2021) with 193 countries aiming to leave no one behind. Among the 17 goals, 10 aim to reduce inequalities, i.e., the reduction of inequalities within and between countries.

According to (Dudás 2019) digitalization achieved through technological innovation contradicts sustainability, (Andersen et al. 2021) because we replace our existing but outdated products, thereby increasing overconsumption (Dudás 2019).

### 2.2. Is Green the New Color of the Financial Sector?

Analyzing consumer behavior is not possible without examining the financial options used during the purchase. A sustainable approach can also be seen in the financial sector.

The reduction of the ecological footprint of the financial sector can be seen in several measures. By 2019, there was a reduction of more than 60% in cash logistics in 8 years, with reusable storage devices being used instead of banknote packaging bags and banknote packaging tapes (Hergár 2020). However, the ecological footprint (e.g., from the decline in paper use to the significant reduction in the number of trips) was also significantly influenced by the external shocks of the past two decades (2008 financial and real economic crisis, fourth industrial revolution, and coronavirus) (Becsei et al. 2021), e.g., the COVID-19 pandemic had a strong impact on the current use of e-wallets (Daragmeh et al. 2021). In addition to all this, other developments have also taken place in the financial sector. At the end of the first quarter of 2021, there were 9.8 million payment cards in circulation in Hungary, which means the use of 49 tons of plastic. Several banks, including the Hungarian OTP Bank, have already issued 85.5% recycled plastic cards, which is a significant development (Portfolio 2021). In many cases, these efforts—carried out by market players—can have a greater impact on the more receptive younger generation, just as the clothing industry does, which is proven to be able to attract consumers to the market by adopting green strategies and implementing environmental protection measures (Dabija 2018).

The previously discussed digitalization also forces the financial sector to change. In Hungary, however, card payments are still not sufficiently developed in smaller settlements and less developed regions (Kajdi and Nemecskó 2020), and there is a clear connection

between the economy's cash flow and the country's innovation performance (Glennow and Granström 2019). According to Geng and He (2021), digital financial inclusion significantly contributes to sustainable employment in high-income economies, especially in upper-middle-income and less significant economies. For lower- and middle-income economies, this effect is insignificant (Geng and He 2021).

The Hungarian Banking Association takes the necessary steps to support the digital transition of the financial sector through active partnership and formulation of proposals (Becsei et al. 2019). At the same time, banking innovation does not only have to be thought of in terms of financial aspects; sociological, psychological, and other economic factors are also important (Csiszárik-Kocsir 2019). It has been proven that those who have already used some kind of technological innovation are more open to electronic money (Hadady 2011). A Malaysian study also found that acceptance of cashless payment methods is greatly influenced by the importance of preparation and anticipation (Rahman et al. 2020). In addition, interesting research has also been published regarding gender. In Kenya, access to mobile payment services has increased by more than 20% among female-headed households. This program, similar in spirit to the Bank of the Poor program, enabled 185,000 women to start a business after giving up agricultural work (Mugo and Kilonzo 2017).

### 2.3. Is the Consumer the Winner of the Battle between Cash and Electronic Money?

In our publication, we analyze the topic of cash payments. We must distinguish the concept of electronic payment from the circulation of electronic money. In Hungary, according to the law, CCXXXVII of 2013, "a sum embodied in a claim against the issuer of electronic money, stored electronically ( . . . ), which is issued against the receipt of funds to complete payment operations specified in the Act on the provision of payment services, and which is made by a natural or legal person other than the issuer of electronic money, it is also accepted by a business company without legal personality and an individual entrepreneur." Electronic money is therefore a future claim that is acquired with current money (cash or transfer). The definition also includes the fact that this electronic money must be accepted by other actors besides the issuer (Banking Act 2013, Section 6 (1) point 16).

The latter condition is affected by the amendment of the 2018 law, according to which the issuance of electronic money, that can only be used in a limited way, is not considered a payment service if: the payment instrument allows it only in the premises used by the issuer or within the private network of service providers that have a direct commercial contract with the issuer, it enables the owner of the device to purchase goods or services, or the payment instrument enables a very limited purchase of goods or services. A payment operation by the operator of the electronic communication network or the provider of the electronic communication service, which is provided in addition to the electronic communication service to the subscriber of the network or service under certain conditions, is also not considered a payment service (Banking Act 2013, Section 6 (4) point l).

The conceptual confusion was increased by the fact that the legal background used the concept of electronic money, in addition to electronic money, meaning a device suitable for storing electronic money. Fortunately, this term is no longer used (Banking Act 2013).

After clarifying the concepts above, the biggest advantages of a cashless society are outlined; a cashless society saves physical resources, eliminates the circulation of wasted or destroyed coins and banknotes during production, saves transportation to and from banks, and saves energy in ATMs. From (Rochemont 2018), by examining the sustainability of smart card payments, it was determined that utility, ease of use, convenience, automated value-added service, security, reliability, and the influence of popular service providers on residential and consumer services (Liao et al. 2014). Based on a population sample of adolescent buyers in Indonesia, the main reasons for using electronic money are practicality, ease of use, efficient transaction time, faster payment, and simplicity of the payment process (Widayat et al. 2020). Moreover, in the SME sector, they found an interesting connection with mobile money in Douala, Cameroon. The most important factors for use were accessibility, safety, and comfort (Tengeh and Gahapa Talom 2020).

However, the process of withdrawing cash is slow. Drehmann et al. (2002), analyzed the payment habits between 1980 and 2000, finding that modern payment techniques have barely left a mark on cash usage habits, and that cash payments are not dying out either. In 2014, many people still did not see the realization of a cashless society in the near future (Bátiz-Lazo et al. 2014), but this can also be traced back to political reasons (Fabris 2019). However, it was established much earlier, in 2006, that the development of a cashless society would probably be economically more beneficial (e.g., for consumers) if not for everyone (certain merchants) (Garcia-Swartz et al. 2006).

During the use of cashless payment transactions, of course, negative aspects also arise, e.g., in the case of Indonesia, data protection (the government will have a complete record of consumers) and security (the emergence of computer hackers) issues have emerged (Ewa Abbas 2017).

In the case of Hungary, there is an interesting trend. According to the Central Bank's report on the Hungarian payment system, in 2019, the proportion of credit card purchases compared to annual household consumption did not reach the EU average but exceeded it by 30% (Hergár 2021). In comparison, a survey conducted in 2020, which examined the impact of the epidemic on food purchases with the participation of 928 people, found that payment by debit/credit card was ahead of cash payment with a share of 58.4%, with cash payments sitting at 37.9%. The rate of mobile payment was 2.2%, while other payment methods (e.g., bank transfer, virtual account) were mentioned in less than 1% (Soós 2020). According to the latest Payment Report, published in 2021, a decisive change in payment transactions took place in Hungary at the end of 2020. According to the report, "it is a significant development ( . . . ) that the number of card purchases exceeded 1 billion, and the value of card purchases exceeded cash withdrawals for the first time" (Hergár 2021).

In terms of the payment card infrastructure, there is significant progress compared to previous years, as shown in Table 1.

**Table 1.** The advancement of touch technology in Hungary.

| Supports Touch Payment | The Magnitude of the Increase Compared to the Previous Year |
| --- | --- |
| Nearly 92% of domestically issued 9.9 million cards | 7% |
| 203,000 POS terminals in operation | 1% |

Table 1 summarizes the state of progress of touch technology at the end of 2020 (Hergár 2021).

### 2.4. Is Financial Awareness Available to Everyone?

Based on the above, consumer behavior and money management are, therefore, related from several points of view. The consumer can spend or even save his/her income, thereby influencing the state of the economy and the environment.

In the case of savings, four consumer groups can be identified according to their preferences. Based on this, the consumer can be risk-averse, conscious, deliberate, or risky (Csernák et al. 2017). Individual consumer behavior patterns can also cause differences. Based on the results of a quantitative study focusing on generations Z and Y, four groups were identified in terms of the price-value ratio as undervaluers, "happy with money", "those who like relationships and savings" and "want to live well and do good" (Garai-Fodor 2018). If someone, has traditionally neglected their finances, choosing to save does not mean that their basic personality will change overnight. However, developing patience and a future-oriented mindset can encourage the consumer to save (Coady 2021). According to a study on financial personality type, frugality and price sensitivity alone are not enough (especially for those prone to impulse purchases), because the most effective strategy is the development of financial awareness (expense planning, accounting) and diligence (Németh et al. 2016).

Educational institutions play a key role in the development of financial culture (Sági et al. 2020). Education and majors influence young people's general and financial product awareness, which can result from assessing the financial risk environment through the acquisition of critical thinking skills (Nga et al. 2010). However, those who start their higher education studies already have to face serious financial constraints (higher education costs are increasing, and the financial possibilities of families are limited). In addition, these factors further strengthen the backwardness (George-Jackson and Gast 2015). On the other hand, according to a survey by Csiszárik-Kocsir and Varga (2018), where financial literacy was examined in the case of young Hungarians entering higher education. The effect of financial socialization proved to be stronger than the financial knowledge acquired in school (Csiszárik-Kocsir and Varga 2018).

The Government of Hungary also recognized the importance of improving the financial awareness of the population and, therefore, created a strategy, one of the goals of which is to improve financial awareness and reduce the use of cash. The completed strategy spans 7 years (2017–2023) and consists of 2-year action plans (Bálint et al. 2020).

Taking the above into account, our article aims to compare the two most common payment methods, cash, and card payment. Based on the literature, it can be concluded that the battle between cash and electronic money is at a turning point, and the tendency is for electronic payments to slowly overcome cash. In light of this, the assessment of consumer behavior concerning the choice of financial instruments is extremely important, since, in the long term, possible disengagement threatens sustainable development goals, and the presence or absence of financial awareness affects the number of purchases and savings. By assessing decision-making factors, we intend to provide a deeper understanding of consumption habits and the spread of digitalization to inform education, which can contribute to the prevention of overconsumption.

Based on the literature, we formulated the following hypotheses:

**H1:** *Decision factors can be defined that influence consumer behavior and the frequency of electronic payment for purchases.*

**H2:** *Payment options that influence financial awareness can reduce the number of purchases.*

### 3. Materials and Methods

#### 3.1. Sampling Method

Based on the literature review, we formulated hypotheses, which were supported by a quantitative survey. The consumer survey ($n = 499$) was conducted online in Hungary between November 2020 and May 2021. The research was conducted with the help of social media. Accordingly, our sampling method was snowball (Goodman 1961), so the respondents were not selected randomly. The advantage of the applied sampling procedure is that it is simple and can be applied to populations that are relatively difficult to access; the disadvantage is that the resulting sample is not representative. The territorial scope was not limited. The survey included a Likert scale (0 = completely disagree; 6 = completely agree), multiple-choice, and open-ended questions about respondent attitudes, payment habits, and personal data (background information on age, education, place of residence, and labor market situation). The questions defined based on the literature were asked in such a way as to ensure the confirmation or refutation of the established hypotheses. The responses were automatically recorded by submitting the completed online questionnaire and then converted into a binary number system for processing.

The distribution of the sociodemographic data of the respondents are shown in Table 2.

**Table 2.** Distribution of the sample.

| Description | | Frequency | % |
|---|---|---|---|
| Age | 16–29 | 147 | 29 |
| | 30–39 | 183 | 37 |
| | 40–49 | 110 | 22 |
| | 50–59 | 45 | 9 |
| | >60 | 14 | 3 |
| Education background | Primary school | 2 | 1 |
| | Secondary school (without graduation) | 27 | 5 |
| | Secondary school (with graduation) | 149 | 30 |
| | College, university | 321 | 64 |
| Residence | Village | 120 | 24 |
| | City | 114 | 23 |
| | County seat | 123 | 25 |
| | Capital city | 142 | 28 |
| Total | | 499 | 100 |

*3.2. Statistical Analysis*

During the processing of the data received, it was necessary to clean the sample, i.e., merge or exclude the categories. Due to the small sample number, the population over 60 ($n = 14$) was combined with the 50–59 age group and renamed as the 50+ category. We did not decide in favor of the exclusion because the thoughts of the older age group were important, and thus the frequency difference between the age groups also decreased. On the other hand, respondents with primary education ($n = 2$) were excluded due to the small sample size, because these two answers were not significant in terms of the obtained results. In the tables presented during the analysis, the data are already presented in this way. The following statements/questions of the questionnaire were used to examine the formulated hypotheses. The individual questions were marked with the numbers "x1-x6"—shown in parentheses—for easier reference and later in the article, we used these notations:

- You will receive your salary in cash. (x1)
- How do you do your shopping regularly? (x2)
- In everyday shopping situations, how gone decide between credit card and cash? (x3)
- Do you spend more where you can pay by credit card, than where you can pay only in cash? (x4)
- I feel safer keeping my money (e.g., my salary) in cash than in electronic form. (x5)
- I think the risk of theft is lower when using a card than when using cash. (x6)

The majority of our chosen questions are Likert scale, the analysis of which does not require complicated, multivariate statistical analysis. In reviewing the literature, it can be seen that some researchers consider the Likert scale as an ordinal variable, while others consider it as an interval measurement level variable. It provides useful information for impact assessments or when examining chronological changes. The relationship between the questions was examined using the Chi-square test. We considered the Likert scale questions as an interval measurement level variable, thus the study compared the independent groups formed according to the given questions of a normally distributed sample based on a common background variable, taking into account the averages of the groups, so we tested the questions with a one-factor ANOVA (Milton 1999). The significance of differences was tested by Tukey's post hoc test and Bonferroni methods. The STATA/MP 15.1 software was used for its analysis.

## 4. Results

We tested our first hypothesis, which attempts to assess the decision factors that influence consumer behavior and the amount of electronic payment for purchases with statements x1-x2-x3.

It is important, from the point of view of the topic and the first hypothesis, whether those who fill in the questionnaire receive their salary in cash, as this can greatly influence the size of the subsequent money usage fees, and convenience can also influence the way the money is spent. According to the results of descriptive statistics, respondents, despite government pressure, still receive their salary in cash (8%, i.e., 40 people), albeit in small numbers. The distribution according to background variables is shown in Table 3, from which it can be established that there is no big difference between the proportions of each category, except for labor market status.

**Table 3.** Results for variable x1 by descriptive statistics.

|  | *n* | % |
|---|---|---|
| Age | | |
| 16–29 | 8 | 20.0 |
| 30–39 | 14 | 35.0 |
| 40–49 | 14 | 35.0 |
| 50–59 | 4 | 10.0 |
| Education background | | |
| Secondary school (without graduation) | 7 | 17.5 |
| Secondary school (with graduation) | 16 | 42.5 |
| College, university | 17 | 42.5 |
| Residence | | |
| Village | 8 | 20.0 |
| City | 11 | 27.5 |
| County seat | 12 | 30.0 |
| Capital city | 9 | 22.5 |
| Labor market status | | |
| Active | 30 | 75.0 |
| Other | 4 | 40.0 |
| Pensioner | 1 | 2.5 |
| Student | 5 | 12.5 |
| Total | 40 | 100 |

In addition to simpler descriptive analyses, cross-tabulation analysis and Chi-square test were also performed to explore deeper correlations. When examining the background variables, in the case of the 40 people, a significant difference occurred only in terms of education (chi-square: 16.639; DF: 3, probability: 0.001).

To make the analysis more accurate, all respondents ($n = 499$) were included in the comparative study. Examining the x1 statement ("You will receive your salary in cash") with the background variables, a significant difference appeared only in the case of education (chi-squared: 16.0588; DF: 3, probability: 0.001). The results obtained are shown in Table 4.

**Table 4.** Results for variable x1 by Cross-table analysis.

| "You Will Receive Your Salary in Cash" | | Qualification | | | Total |
| | | College, University | Secondary School (with Grad.) | Secondary School (with Grad.) | |
|---|---|---|---|---|---|
| Yes | Count | 17 a | 7 b | 15 a,b | 39 |
| | % within payment in cash | 43.6 | 17.9 | 38.5% | 100.0 |
| No | Count | 303 a | 20 b | 133 a,b | 456 |
| | % within payment in cash | 66.4 | 4.4 | 29.2 | 100.0 |
| Total | Count | 320 | 27 | 148 | 495 |
| | % within payment in cash | 64.6 | 5.5 | 29.9 | 100.0 |

Each subscript letter denotes a subset of qualification categories whose column proportions do not differ significantly from each other at the 0.05 level.

Following the cross-tabulation analysis, we used the Bonferroni test to examine where there was a justified difference between the qualification categories and the choice of payment method. After running it, we obtained the result that there was a justified difference between the choice of each qualification category and the method of payment in all cases, except for the salary received in the bank account and those with secondary education without a high school diploma. Thus, education indirectly influences the choice of payment method and the nature of the job while the job determines who gets paid.

The second examined statement (x2) focused on a regularly used means of payment for managing purchases ("How do you regularly manage your purchases?"). Based. on the results, a relatively large number—68% of respondents—prefer to make their daily purchases with a debit card.

From the statistical studies (cross-tabulation analysis, Chi-square test) it can be stated that the difference confirmed during the analysis of correlations with the background variables was also shown in the case of education (chi-squared: 42.336; DF: 6, probability: 0.000). During the running of the Bonferroni test, we obtained a similar result as in the previous statement of x1; there was no significant difference between the individual qualification categories (Table 5).

**Table 5.** Results for variable x2: significant difference between categories by Chi-square test.

| "How Do You Do Your Shopping Regularly?" | | Qualification | | | Total |
| | | College, University | Secondary School (without Grad.) | Secondary School (with Grad.) | |
|---|---|---|---|---|---|
| would rather by card | Count | 248 a | 8 b | 84 c | 340 |
| | % within shopping | 72.9% | 2.4% | 24.7% | 100.0% |
| would rather in cash | Count | 17 a | 5 b | 20 b | 42 |
| | % within shopping | 40.5% | 11.9% | 47.6% | 100.0% |
| Combining | Count | 56 a | 14 b | 45 b | 115 |
| | % within shopping | 48.7% | 12.2% | 39.1% | 100.0% |
| Total | Count | 321 | 27 | 149 | 497 |
| | % within shopping | 64.6% | 5.4% | 30.0% | 100.0% |

Each subscript letter denotes a subset of Qualification categories whose column proportions do not differ significantly from each other at the 0.05 level.

Statement x3 provides information on the factors we used to choose between payment instruments in everyday shopping situations ("In everyday shopping situations, what makes you decide between a credit card and cash?"). Respondents had to evaluate their choice of payment methods on a Likert scale, but we also provided an open-ended opportunity to express their views. In total, 44.3% of respondents said they did not make decisions based on habit, while 29.5% were influenced by everyday situations. As before, we disaggregated the analysis by age, education, labor market status, and place of residence to explore deeper correlations by the ANOVA method. In the decision to choose a means of payment, a significant difference can be seen for the location of payment and habit, as shown in Table 6.

**Table 6.** Significant factors for deciding on the choice of the payment method.

| | Capital City | | | City | | | County Seat | | | Village | | | *p*-Value |
|---|---|---|---|---|---|---|---|---|---|---|---|---|---|
| | Mean | Sd | Sem | Mean | Sd | Sem | Mean | Sd | Sem | Mean | Sd | Sem | |
| The volume of the amount to be paid | 1.88 | 2.02 | 0.17 | 2.51 | 1.86 | 0.18 | 2.08 | 1.94 | 0.18 | 2.29 | 1.96 | 0.18 | 0.075 |
| Payment location | 2.87 | 1.98 | 0.17 | 3.06 | 1.82 | 0.17 | 2.34 | 1.95 | 0.18 | 2.61 | 1.87 | 0.17 | 0.024 |
| Habit | 2.2 | 2.03 | 0.17 | 2.18 | 1.94 | 0.19 | 1.68 | 1.84 | 0.17 | 2.35 | 1.92 | 0.18 | 0.050 |
| Other | 1.85 | 2.14 | 0.26 | 1.74 | 2.04 | 0.33 | 2.39 | 2.02 | 0.32 | 1.76 | 2.10 | 0.31 | 0.439 |

The results for the analysis of variance show that there is a statistically verifiable difference between the location of the payment, the decision factors of habit, and the place of residence. A Tukey test was performed to detect differences between groups, the results of which are illustrated in Table 7.

**Table 7.** Tukey test for x3 variable grouping by residence.

| Decision Factor | Pairwise | Mean Diff. | Std. Error | Sig. | 95% Confidence Interval | |
|---|---|---|---|---|---|---|
| | | | | | Lower Bound | Upper Bound |
| Location of payment | 2–4 | 0.725 * | 0.253 | 0.023 | 0.07 | 1.38 |
| Habit | 3–4 | 0.668 * | 0.256 | 0.047 | 0.01 | 1.33 |

Notes: 2: City; 3: Village; 4: County seat. * significant difference between the variables.

In the current study, there is a justified difference between the village and the county seat, in terms of the location of the payment, the custom of the other town, and the county seats.

In the case of the other option (open-ended question), 117 possible answers were returned, of which 114 interpretable answers were accepted after processing, where financial awareness appears significantly (Figure 1). The number of evaluable responses received was 114, and one person's response could be classified into more than one category. Accordingly, at 50, an external factor determines the payment method, e.g., the supply of a given point of sale with a terminal, or how a given person obtains the income. In total, 21 make ad-hoc decisions, while 43 deliberately vote for cash or a card for economic, convenience, or transparency reasons. These categories have been defined by the authors on their own merits based on the literature.

Overall, the statistical analysis results show that in the case of x1 (chi-square: 16.0588; DF: 3, probability: 0.001) and x2 (chi-square: 42.336. DF: 6, probability: 0.000) there is a significant difference with the background variables only in the case of educational attainment. However, in the case of x3, there is a difference between the place of residence, including the village and the county seat.

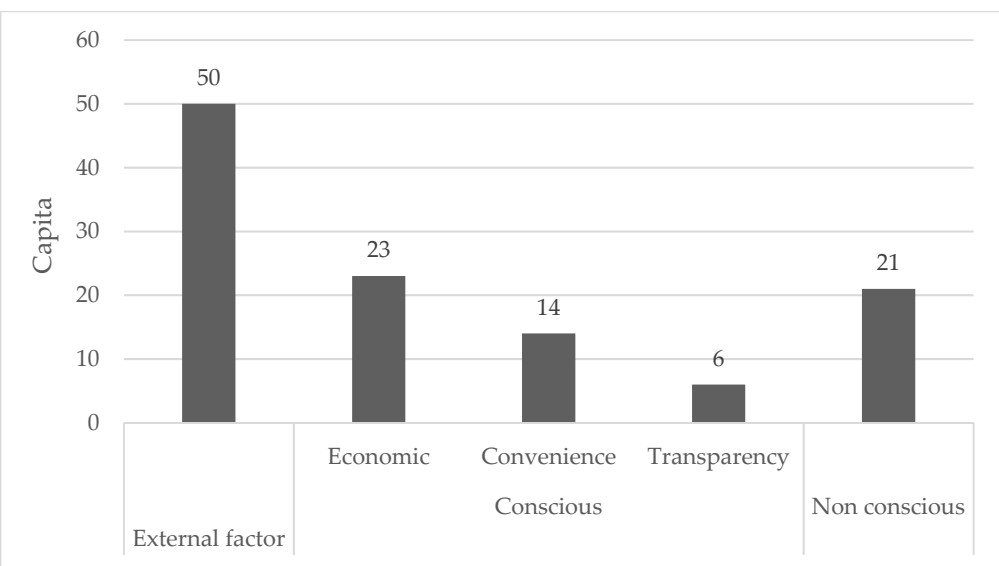

**Figure 1.** Subjective responses to the payment method decision (*n* = 114).

Based on the above, we accepted our first hypothesis ("Decision factors can be defined that influence consumer behavior and the frequency of electronic payment for purchases").

Thus, based on the results, consumers did not use cash and card uniformly in their various payments yet, but it may be interesting to examine what motivates them to make a choice and how conscious the outcome is.

We tested our second hypothesis with statements x4-x5-x6.

With statement x4, we aimed to examine whether respondents' shopping habits are affected by the payment instrument they choose, which can generate overspending, jeopardizing sustainability ("Where you can pay by credit card, you spend more than you can with cash only.").

The descriptive statistics showed that 18.24% of respondents do not know or did not observe their spending habits, of those who could tell 17.23% knew they were spending more and 64.53% said they were not spending more when paying by card.

There is no statistically significant difference with background variables, which means that statement x4 is independent of age, place of residence, education, and labor market status.

Concerning the issue, it is important to highlight that financial awareness did not occur in nearly a fifth of respondents because they did not know or did not pay attention to their spending habits. Awareness is an important part of managing our finances, the existence of which contributes to long-term sustainability.

Another question of the research confirms this result, where we asked an open-ended question about the most important features and advantages of cash payment. In total, 20 of the 26 evaluable responses were returned, of which awareness was most often mentioned (Figure 2). Several respondents have written that traceability (eight people) is an important advantage in choosing the form of payment, five have emphasized the economic aspect, and four consider the use of cash to be more convenient.

The x5 and x6 statements provide information on the attitudes of those surveyed towards security ("I feel safer keeping my money (e.g., my salary) in cash than in electronic form" and, "I think the risk of theft is lower when using a card than when using cash", respectively).

Based on the data, it can be stated that in the case of variable x5, we found a significant difference based only on education and employment status. In the case of theft, in relation to variable x6, we did not experience a significant difference between the individual background variables, as shown in Table 8.

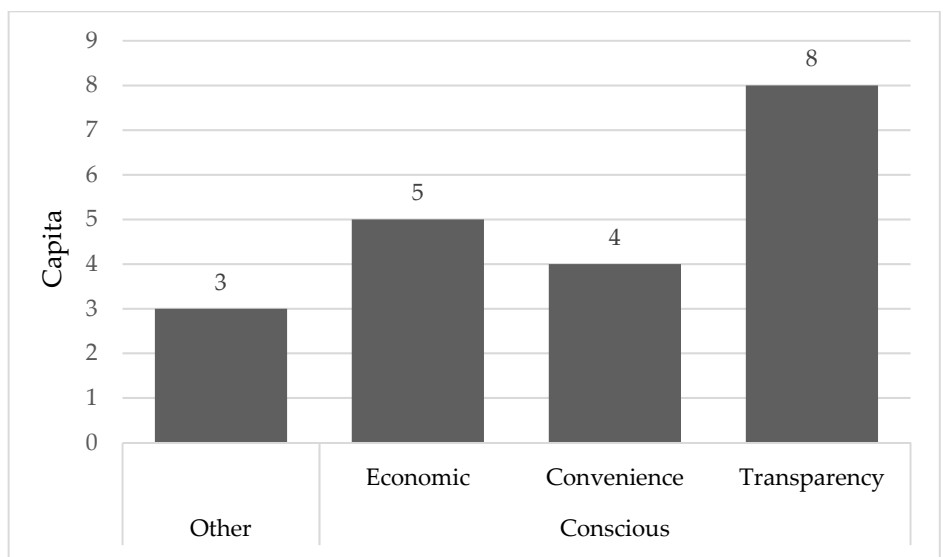

**Figure 2.** The benefits of cash payment based on respondents (*n* = 20).

**Table 8.** Results for variable x5-x6 by Chi-square test.

| | "I Feel Safer to Keep My Money (e.g., My Salary) in Cash than in Electronic Form" | | | "I Think the Risk of Theft Is Lower When Using a Card than When Using Cash" | | |
|---|---|---|---|---|---|---|
| | Mean | Sd | *p*-Value | Mean | Sd | *p*-Value |
| Age | | | | | | |
| 16–29 | 1.37 | 1.724 | | 3.37 | 1.33 | |
| 30–39 | 1.21 | 1.468 | 0.497 | 3.27 | 1.41 | 0.167 |
| 40–49 | 1.50 | 1.696 | | 2.96 | 1.477 | |
| >50 | 1.56 | 1.787 | | 3.19 | 1.454 | |
| Education background | | | | | | |
| Secondary school (without graduation) | 2.41 | 1.845 | | 3.30 | 1.564 | |
| Secondary school (with graduation) | 1.74 | 1.828 | 0.000 | 3.23 | 1.468 | 0.61 |
| College, university | 1.07 | 1.413 | | 3.19 | 1.379 | |
| Residence | | | | | | |
| Village | 1.58 | 1.713 | | 3.13 | 1.507 | |
| City | 1.39 | 1.681 | 0.197 | 3.25 | 1.436 | 0.651 |
| County seat | 1.31 | 1.595 | | 3.13 | 1.379 | |
| Capital city | 1.15 | 1.502 | | 3.32 | 1.354 | |
| Labor market status | | | | | | |
| Active | 1.26 | 1.552 | | 3.18 | 1.44 | |
| Other | 1.38 | 1.601 | | 3.14 | 1.39 | |
| Unemployed | 0 | 0 | 0.005 | 3.33 | 1.528 | 0.587 |
| Pensioner | 2.08 | 2.021 | | 3.25 | 1.357 | |
| Student | 2.37 | 2.166 | | 3.53 | 1.020 | |

In Table 9, the segregation relationships between the x5 variable and the background variables are highlighted. Based on our research, respondents who do not have a high school diploma are different from those with a high school diploma, and those with higher education, prefer to use cash. In addition, there is a difference based on labor market status, where students differ from all other categories, i.e., students feel more secure using cash. This may be because students spend smaller sums as they mostly do not have an independent income, and for smaller amounts, only cash is available in several locations.

**Table 9.** Tukey test for x5 variable grouping by education background and labor market status.

| Decision Factor | Pairwise | Mean Diff. | Std. Error | Sig. | 95% Confidence Interval | |
|---|---|---|---|---|---|---|
| | | | | | Lower Bound | Upper Bound |
| Education background | 2–3 | 1.339 | 0.316 | 0.001 | 2.15 | 0.52 |
| | 2–4 | 0.676 | 0.156 | 0.001 | 1.08 | 0.27 |
| Labor market status | 1–5 | 1.106 | 0.337 | 0.040 | 2.18 | 0.03 |

Notes: Educational background: 2: College, Univesity; 3: Secondary school (without graduation), 4: Secondary school (with graduation); Labor market status: 1: Active, 5: Student.

Based on the x4 statement, 18.24% do not know or follow their spending habits, 17.23% knew they would spend more, and 64.53% said they would not spend more on card payments. Based on the data, it can be stated that in the case of variable x5, we found a significant difference based only on education and employment status. In the case of theft and variable x6, we did not experience a significant difference between the individual background variables.

Based on the above, we rejected Hypothesis H2 ("Payment options that influence financial awareness can reduce the number of purchases."), the degree of financial awareness does not affect the choice of payment method.

## 5. Conclusions

Considering the above, our article aims to compare the two most common payment methods, cash, and card payments. Based on the literature, it can be concluded that the battle between cash and electronic money is at a turning point and there is a tendency for electronic payments to slowly overcome cash.

In light of this, the assessment of consumer behavior regarding the choice of financial instruments may be extremely important shortly. On one hand, this possible separation threatens sustainable development goals in the long term, and on the other hand, financial awareness can affect the number of purchases and savings.

Based on our literature research, we assumed that there were decision factors that influenced payment habits. Age (Éltető 2021) and educational background (Nga et al. 2010) can be derived directly from the literature, which indirectly affects the labor market situation and place of residence. Based on the tests carried out, we have found that our results, in terms of education and labor market status, agree with the literature. In addition, the location of payment and habit as a decision-making factor demonstrably influence consumer behavior and the chosen payment method. Based on the above, we accept our first hypothesis ("Decision factors can be defined that influence consumer behavior and the frequency of electronic payment for purchases").

By assessing decision-making factors, we intend to provide a deeper understanding of consumption habits and the spread of digitalization to inform education, which can contribute to the prevention of overconsumption. In our second hypothesis, we examined whether financial awareness reduced the number of purchases (Helm and Subramaniam 2019). Based on the results of our research, the full transition is not yet feasible, as shown by the results of our survey. Eight percent of respondents still receive payment in cash, 8.4% still buy only in cash, and 31.7% use cash and cards alternately when shopping. Based on the results we can conclude that financial awareness is not affected by the choice of payment method, which is why we reject our second hypothesis. Although government pressure would force cashlessness (thereby explicitly whitening the economy), switching opportunities are still not available in many places (e.g., smaller beauty providers, markets, etc.).

For sustainability, the UN aims to eliminate inequalities. As financial awareness is strongly influenced by education, according to the literature and according to our research. Therefore, the lagging strata are at multiple disadvantages. In total, 17.23% of our re-

spondents know that they spend more on card payments, and 29.5% are influenced by their habits in everyday shopping situations. This "spontaneity" in the choice of payment methods refers to a lack of awareness, including financial awareness, which increases consumption and, thus, the amount of waste, jeopardizing sustainability.

In the fight against the use of cash, the introduction of digital central bank money is expected to have a decisive influence in the near future. Based on this, the central bank maintains a special account for all actors, the use of which is expected to result in a significant reduction in cash, since digital central bank money will function as a kind of digital cash. In our study, we could not address this expected change, this direction may mean the continuation of our research.

**Author Contributions:** Conceptualization, Z.P.; Methodology, M.Z.N.; Supervision, J.V.; Writing—original draft, Z.P. and K.T.; Writing—review and editing, J.V. and M.Z.N.; Formal analysis, Z.P.; Administration, K.T. All authors have read and agreed to the published version of the manuscript.

**Funding:** This research received no external funding.

**Informed Consent Statement:** Informed consent was obtained from all subjects involved in the study.

**Data Availability Statement:** The data presented in this study are available on request from the authors.

**Acknowledgments:** Special thanks to Erzsébet Kopházi-Molnár for reviewing and correcting the manuscript.

**Conflicts of Interest:** The authors declare no conflict of interest.

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
