# Peer review of "The Struggle between Cash and Electronic Payments"

_economies, doi:10.3390/economies10120304_

Round 1
Reviewer 1 Report (Previous Reviewer 2)
Comments to Authors
When I carefully checked the Authors' point-to-point replies to my suggestions, I noticed that the Authors had addressed my recommendations in my review. The authors have redistributed the manuscript to the Literature review with an introduction. However, I suggest the authors further improve the introduction section of the manuscript, as the revised version still has some issues. In the introduction section, Author/s can provide the research background for the Struggle Between Cash and Electronic Payments, the specific research problem, objectives, the significance of the study, and an overview of the paper's structure, etc., to enhance the manuscript's readability. Authors may discuss the main points in sub-sections. Also, I suggest the Authors include more relevant and latest scientific/classic references.
Further comments to Authors
I believe extensive English language and style editing will strengthen the manuscript's academic sound. I recommend this research paper for publication with minor corrections.
All the best!
Author Response
Please find the attached file.

Reviewer 2 Report (Previous Reviewer 1)
The author has properly replied to my previous concern. I have no more questions.
Author Response
Please find the attached file.

This manuscript is a resubmission of an earlier submission. The following is a list of the peer review reports and author responses from that submission.
Round 1
Reviewer 1 Report
This paper reports a survey on the impact of payment methods on financial awareness. The topic is very interesting and is worthy for research, however, there are some concerns on this study.
1. The subsection of the introduction is less connected. Three parts seem unrelated to each other. The author should focus their research question and re-organize the intro.
2. The statistical analysis seems ambiguous. Why the authot choose certain weights? Also, the sampling should be carefully discussed.
3. In the ANOVA equation, the label [45] is misleading.
4. The result part seems overwritten. Some of the tables can be merged to discuss.
Reviewer 2 Report
Comments to Author/s
The core of this research paper is good. The author/s took up the issue of the struggle between cash and electronic payments, which fully accords with the Economies journal topics. The article examines the reasons for the decisions, mainly focussing on financial awareness. It compares the common payment methods and cash and card payments using 499 Hungarian people. The paper closely corresponds to the topic specified in the title.
Though this paper has some evident strengths, especially with the topic’s attractiveness, some elements need improvement. I recommend this research paper for publication with a few corrections since this manuscript has some limitations.
I suggest the following adjustments to this article:
1. Author/s must further improve the Abstract by incorporating the study’s critical findings and the key policy recommendations. It would be highly appreciated if the authors could provide the results in words and let the data analysis and discussion sections give the details. Preferably, the Abstract should not include descriptive statistics.
2. It is better if the manuscript presents the study background and literature review as separate sections. Hence, I suggest the authors divide the Introduction into sections: Section One – Introduction and Section Two – the Literature review. In the introduction section, you can provide the research background, the specific research problem, specific objectives, and an overview of the paper’s structure to enhance the manuscript’s readability.
3. The Materials and Methods section shows a very high similarity score. Hence, please recheck the writing in this section. I hope the authors will reduce similarities with the publication of Pintér et al. 2021.
4. The methodology is slightly flawed because it lacks sufficient justification for choosing a 499 sample in snowball sampling methods for data collection. The sample may only represent a layer of society. Please justify your sampling method to generalize the key finding for Hungarian society.
5. The consumer survey was held between November 2020 and May 2021. To my awareness, this was the time of the Covid-19 pandemic, a particular period and context. However, Author/s did not consider the COVID-19 impact on consumer decisions regarding payment methods. Most people’s behavior changed during COVID-19, and some literature shows a significant COVID-19 effect on consumers’ payment behavior. The materials and method section should justify this or discuss it as a study limitation.
6. In section 2.2 of statistical analysis, the Author mentioned that the questionnaire was evaluated with One-way ANOVA, and the f-test and t-test verified the proven differences in variance and means. As ANOVA and t-tests are parametric tools, authors should justify the techniques and their fundamental assumptions.
7. Results indicate that many respondents prefer to make daily purchases with debit cards. Also, it explains that most Hungarians are not yet ready to accept e-payments in full. Please justify and discuss this point with previous literature.
8. It would be highly appreciated if the authors could mention the statistical tools used for the results produced in the tables.
9. Based on the results, the research accepts the first hypothesis and rejects the second hypothesis (Financial awareness does not affect the choice of payment methods). Please discuss these points with past literature, and discuss the critical policy recommendations suggested by the authors for sustainability.
Thank you for submitting this paper. I hope the author/s find(s) these suggestions helpful in improving the contribution to the literature made by this paper. All the best!